# Modeling the SARS-CoV-2 epidemic and the efficacy of different vaccines across different network structures

**Gregg Hartvigsen**[ID]*, **Yannis Dimitroff**‡

Biology Department, SUNY College at Geneseo, Geneseo, New York, United States of America

‡ Current Address: SUNY Upstate Medical University, Syracuse, New York, United States of America
* hartvig@geneseo.edu

**Data availability statement:** All R computer code files are available from the GitHub database

## Abstract

We developed a network-based SEIRV model to test different vaccine efficacies on SARS-CoV-2 (*Betacoronavirus pandemicum*) dynamics in a naive population of 25,000 susceptible adults. Different vaccine efficacies, derived from data, were administered at different rates across a range of different Watts-Strogatz network structures. The model suggests that differences among vaccines were of minor importance compared to vaccination rates and network structure. Additionally, we tested the effect of strain differences in transmissibility ($R_0$ values of 2.5 and 5.0) and found that this was the most important factor influencing the number of individuals ultimately infected. However, network structure was most important in affecting the maximum number of individuals that were infectious during the epidemic peak. The interaction of network structure, vaccination effort, and difference in strain transmissibility was highly significant for all epidemic metrics. The model suggests that differences in vaccine efficacy are not as important as vaccination rate in reducing epidemic sizes. Further, the importance of the evolution of viral transmission rates and our ability to develop effective vaccines to combat these strains will be of primary concern for our ability to control future disease epidemics.

## 1 Introduction

Severe acute respiratory syndrome coronavirus 2 (SARS-CoV-2) began spreading through the human population at the end of 2019. The virus is highly infectious and continues to spread through the population after having caused widespread morbidity, mortality, and disruptions of basic human activities worldwide [1]. A multinational effort with multiple pharmaceutical companies led to the production of several vaccines. An additional challenge was that vaccines were not made available equitably across nations. The challenge remains as to determine the efficacy of different vaccines with multiple doses to increase immune response. Additionally, vaccine hesitancy remains a challenge nearly everywhere [2].

The different SARS-CoV-2 vaccines exhibit different efficacies in stimulating immunity [3]. Additionally, data suggest that individuals respond differently to these vaccines but that all vaccines reduce the levels of morbidity, mortality, and the rates of hospitalizations [4–6]. Various clinical trials also have differed in their target audiences, which helps to explain why

(https://github.com/GreggHartvigsen/
Modeling-COVID-Vaccine-Efficacy). and at
Dryad:
https://doi.org/10.5061/dryad.98sf7m0vh.

**Funding:** The author(s) received no specific
funding for this work.

efficacies vary [3]. Additionally, effectiveness varies among patients based on underlying health risks [7]. Unfortunately, clinical trials also usually only assess disease expression rather than the likelihood that individuals can spread the disease agent [8].

An important question remains as to whether vaccines for SARS-CoV-2 or other disease agents, with their different efficacies, lead to different rates of morbidity, especially when populations may be exhibiting different levels of connectivity (e.g., visiting only with family members or interacting with individuals throughout a community). It has been shown that different efficacies of a first vaccine dose may be of less importance if individuals receive a second dose of a vaccine [9]. Additionally, reducing transmission can lead to an increase in virulence by favoring agents with higher growth rates [10]. In this paper, we test the effects of employing vaccines with a range of efficacies that were found in the range of the early vaccines that were administered and test the effectiveness of one dose versus the addition of a second booster dose. However, to match the fact that vaccines are imperfect, we assume vaccine efficacy never reaches 100% [11]. Additionally, it's important to note that we assume vaccinated individuals are less likely to transmit the virus to neighbors, that all individuals have equal responses to vaccines, and that immune responses to infection are homogeneous.

During the early days of the SARS-CoV-2, daily vaccine rates in the United States ranged from 1-4 million individuals per day, or approximated 0.301 to 1.205% per day, assuming a population of 332 million individuals. Unfortunately, early stages of vaccinations in most countries (e.g., USA and England) withheld vaccinating young individuals who also are more likely to be asymptomatic spreaders [12]. More troubling, however, is that the daily delivery of doses declined [13]. We, therefore, investigate the effects of vaccinating populations over a range of daily delivery rates.

In this paper we simulate the emergence of a SARS-CoV-2-like virus in a naive population and test how deploying vaccines with different efficacies, and the rate of deployment of either one or two doses, affects disease spread. We chose to use two different strains, having reproductive rates ($R_0$) of 2.5 and 5.0. These values represent the average number of secondary infections that would occur from a single infectious individual in a fully susceptible population (see Methods). These values are relatively conservative, with median values having been estimated for SARS-CoV-2 to range from 3.5 to 5.9 across different countries, including the United States [14]. Additionally, we investigate several response variables that assess the extent of the epidemic. These include the number of individuals infected, the duration of the epidemic, the maximum number of individuals infectious (epidemic peak), and the day this maximum occurs. Reducing the epidemic peak size, often referred to as working to "flatten the curve," is critical for health care systems to be able to accommodate the affected population before they become overloaded.

The primary goal of this work is to help us gain greater insight into the emergence of new viral disease agents and what responses can be implemented so as to reduce the rates of morbidity and to help inform vaccine deployment under a variety of conditions.

## 2 Methods

We developed an SEIRV epidemiological model (Fig 1) that runs on a Watts-Strogatz small-world network [15] with 25,000 individuals (see Fig 1 for the definition of these states). These networks are first constructed as circular networks with each individual connected to the five nearest individuals to the left and five to the right ($k = 10$). Then, for each individual, each of their edges to their neighbors is detached from their neighbor with probability $P$ and connected to a randomly chosen individual (except self) in the population. We chose five different values for $P$, resulting in different network structures, ranging from nearly structured

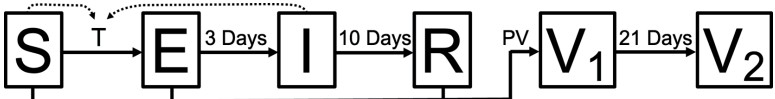

**Fig 1. The structure of the SEIRV model.**

as a regular circulant ($P$ = 0.004) to a relatively random network ($P$ = 0.5, see Table 1). These particular network structures were chosen based on the responses of disease dynamics across the range of possible networks without vaccination (see Fig 2). We chose the Watts-Strogatz structure for its properties that allow us to test quite different networks while maintaining a constant number of edges and, therefore, an identical median degree ($k$ = 10). Our model assumes that all individuals are equally susceptible to viral infection. However, we recognize that individuals may gain some protection from prior exposures, such as to the endemic human coronaviruses [16] but here, assume that no individuals have previously been infected. Each run began by creating a new small-world network with one of five rewiring parameter values (see Table 1). At the beginning of each simulation, six randomly chosen individuals were inoculated with an agent simulating the SARS-CoV-2 virus with three individuals being in the exposed class ($E$) and three in the infectious class ($I$).

Compartments represent states of individuals, being either susceptible ($S$), exposed ($E$), infectious ($I$), recovered ($R$), or individuals that have received the first ($V_1$) or second ($V_2$) dose of the vaccine. Solid arrows indicate the possible movement of individuals from one compartment to another. Susceptible individuals move into the exposed class with probability $T$ for each of their neighbors that are infectious (dashed arrows; see Eq. 1). Unvaccinated individuals in the $S$, $E$, or $R$ states are randomly chosen to be vaccinated with probability PV. Individuals move deterministically from $E \rightarrow I$, $I \rightarrow R$, and from $V_1 \rightarrow V_2$.

Network structure is controlled by the parameter $P$, which is the proportion of edges that are randomly chosen, disconnected, and then reconnected to a randomly chosen individual in the population (see [15]). The red vertical bars represent the five network structures used

**Table 1. Parameter settings for simulations. This table contains the different values tested, resulting in a total of 64,000 simulations. The percent of the population vaccinated daily is an upper limit since only S, E, and R individuals could be vaccinated. Maximum efficacy refers to the probability that a vaccinated person is protected from getting infected 21 days after receiving the vaccine.**

| Parameter | Settings |
|---|---|
| N | 25,000 |
| $R_0$ | 2.5, 5.0 |
| Initial number E | 3 |
| Initial number I | 3 |
| Days E ($D_E$) | 3 |
| Days I ($D_I$) | 10 |
| Average number of neighbors ($k$) | 10 |
| Network rewiring probability ($P$) | 0.004, 0.01, 0.03, 0.1, 0.5 |
| Vaccination strategies (VS) | random, high degree |
| % vaccinated day$^{-1}$ (% Vacc/Day) | 0, 0.3%, 0.6%, 0.9%, 1.2% |
| Number of doses individuals received (ND) | 1, 2 |
| Maximum efficacy (probability) of dose #1 ($V_{max,1}$) | 0.4, 0.45, 0.5, 0.55, 0.6, 0.65, 0.7, 0.75 |
| Maximum efficacy (probability) of dose #2 ($V_{max,2}$) | 0.75, 0.80, 0.85, 0.9 |
| Number replicates | 10 |

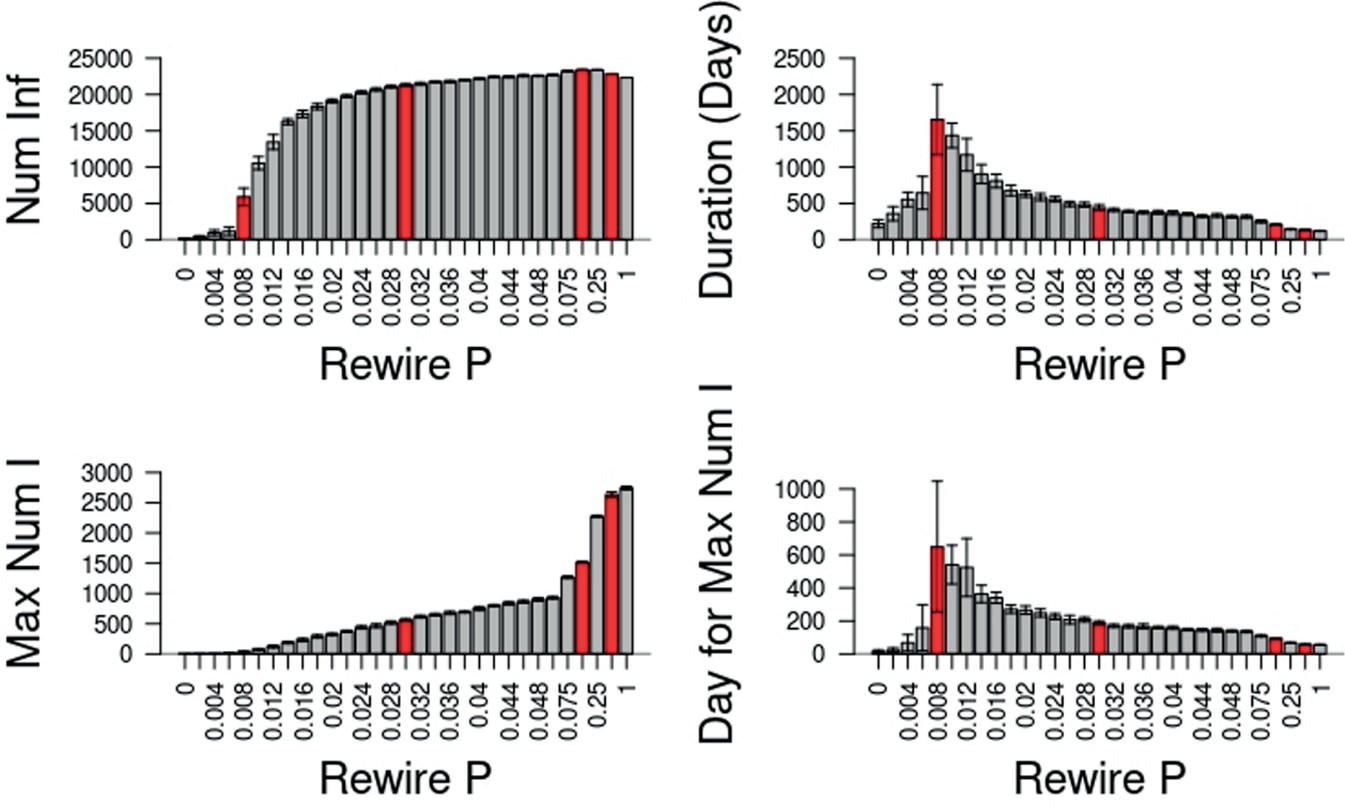

**Fig 2. Model sensitivity to network structure.**

in the simulations (see Table 1). Note the responses for the total number of infected individuals (upper left) rise quickly at about $P = 0.004$. Additionally, average epidemic duration (upper right) and the average day that the epidemic has the highest prevalence (lower right) are greatest at this level of rewiring. All simulations were run with $N = 25,000$ and $k = 10$. Each setting of the network structure parameter $P$ was replicated 10 times in the absence of vaccination. Error bars represent ± 95% CI.

The advantage of this family of networks is that we can simulate population structures across a range of clustering coefficients and average path lengths using the single parameter $P$ while holding the number of vertices and edges constant. Therefore, responses we see with different values of $P$ are due to how individuals are connected in the network. Edges in these networks are undirected and represent possible transmission routes for the SARS-CoV-2 virus among individuals. For comparison, we can see that the degree distributions of the five networks are quite different (Fig 3).

Sample degree distributions for the small world network structures used in this model. Rewiring probabilities ($P$) are shown and match those that were tested in Fig 2. Simulations were run with $N = 25,000$ and $k = 10$. Sample networks show the structures with 250 vertices. Note that a new network was generated for each of the 64,000 simulations.

All individuals were equally susceptible to contracting the SARS-CoV-2 virus from an infectious neighbor. The transmission probability ($T$) from an infectious individual to a

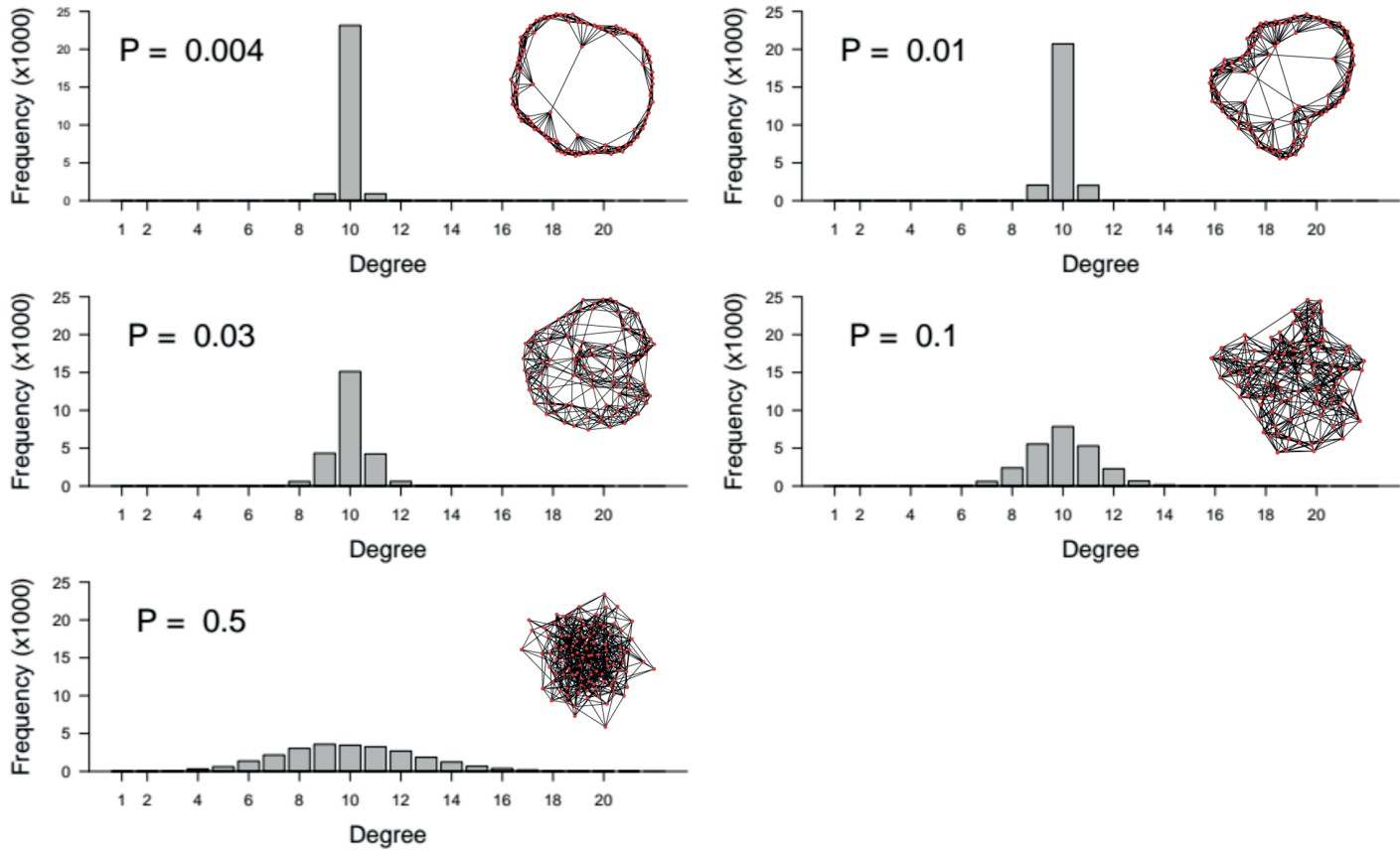

**Fig 3. Degree distributions.**

susceptible neighbor on a single day was determined using the following relationship [17].

$$T = 1 - \left(1 - \frac{R_0}{k}\right)^{1/D_I}$$

(1)

where $R_0$ is the average number of secondary infections caused by an infectious individual in a fully susceptible neighborhood, $k$ is the mean degree of the network, and $D_I$ is the number of days an individual is infectious [18] (see Table 1). In this model we tested the effects of using $R_0$ values of 2.5 and 5.0. This relationship for $T$ results in infectious individuals, on average, infecting $R_0$ susceptible neighbors in a completely susceptible neighborhood. However, as the infection spreads the realized spread rate decreases as the number of susceptible neighbors of infectious individuals decreases. In the event that a susceptible, but vaccinated, individual is chosen for infection by an infectious neighbor, the probability that they become infected is determined by the efficacy level, calculated using Eq. 2 (see below). With time, vaccinated individuals become less susceptible to infection (see Fig 4).

The efficacy of vaccines tested (black lines) when administered on day 1. Efficacy in this model is the probability that the individual will not become infected when challenged by an infectious neighbor. We assume that efficacies increase sigmoidally such that it takes time for the immune system to respond (Eq. 2). The estimated values of efficacies for four vaccines (colored) are shown in comparison to those tested in the model (black lines). The Johnson

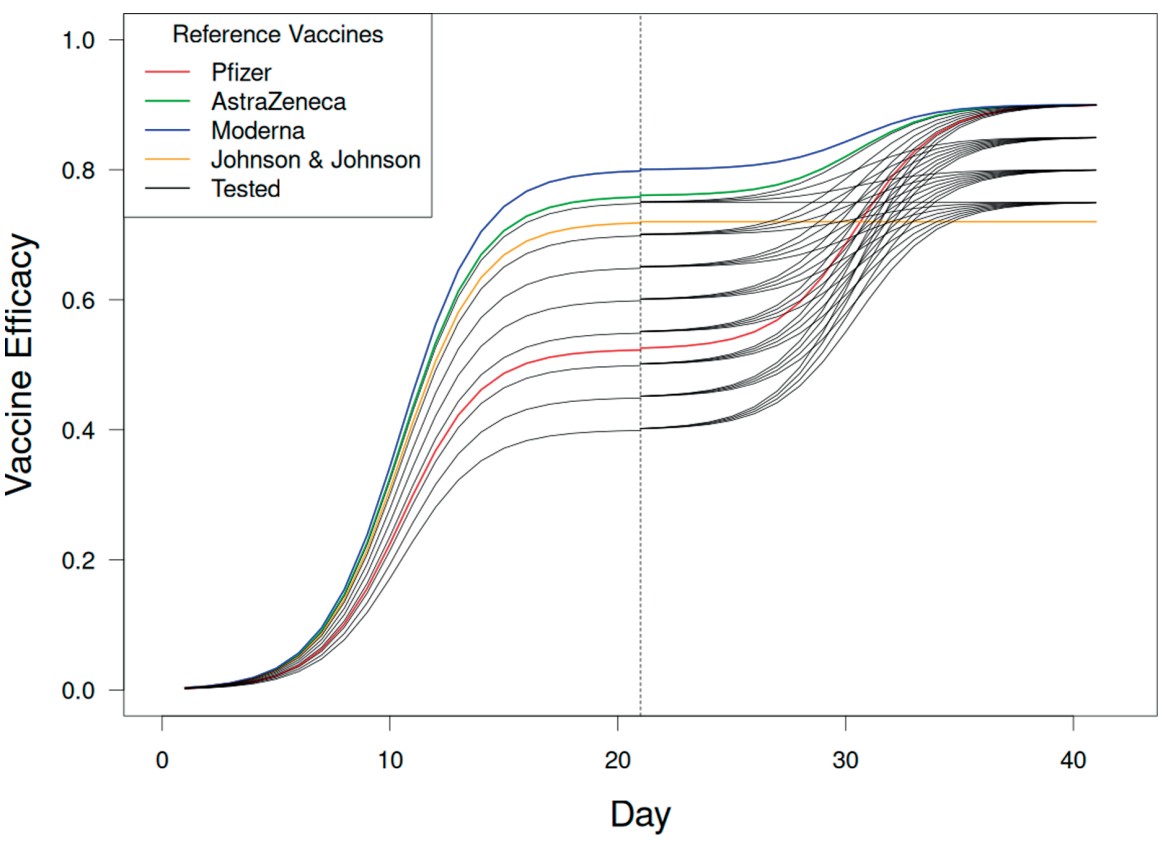

**Fig 4. Change in vaccine efficacies over time.**

& Johnson vaccine (JJ) required only one dose and exhibits 72% efficacy. The remaining reference and tested vaccines were administered 21 days after receiving the first vaccine dose (vertical dashed line). Note AZ represents the AstraZenca vaccine.

## Assessing how $R_0$ affects infection rates in the absence of vaccination

Disease agents, like SARS-CoV-2, typically spread through populations at different rates [19]. Although difficult to estimate, the parameter $R_0$ is used to describe the approximate number of secondary infections that a single infectious individual would cause in an otherwise naive population. In a differential equation model with a fully connected population an $R_0$ value of 1.0 would lead to a sustained number of infectious individuals while values greater than one lead to an epidemic. This is not true in a realistically-structured network population, as explored in this paper, which requires $R_0$ values greater than about 1.2 for an epidemic to emerge. This, however, is still highly dependent on the structure of the network (Fig 5). As a network becomes more randomly connected (increasing values of $P$) both the clustering coefficient and the average path length decrease, allowing the pathogen to more quickly and widely spread through a population.

The size of the epidemic (number of individuals that became infected) is influenced by both $R_0$ and network structure ($P$). Large epidemics are more likely to occur with larger values of $R_0$. On relatively regular networks ($P<0.03$), epidemics only occur when $R_0$ is well above 1.0. The model was run with $N = 25,000$, $k = 10$, and no vaccination. The blue and red

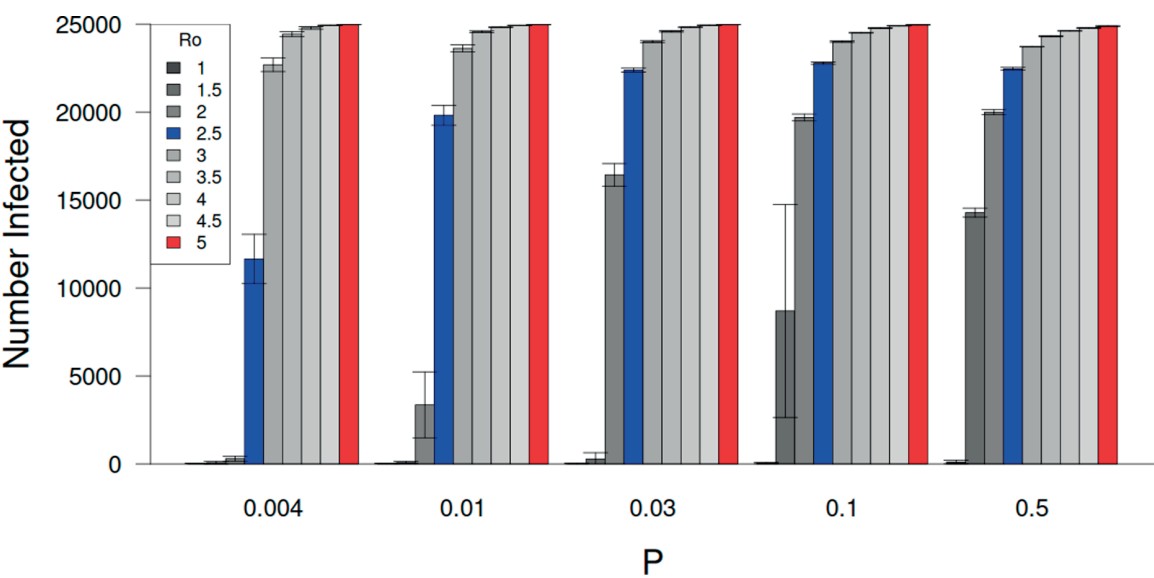

**Fig 5. Epidemic sizes for different $R_0$ values and network structures.**

bars represent the $R_0$ values tested (2.5 and 5.0, respectively). Error bars represent $\pm$ 95% confidence intervals.

### Implementing vaccinations

We tested the effectiveness of vaccines that exhibit different levels of efficacy, ranging from 40% to 75% protection for the first dose and from 75% to 90% with the second dose (see Table 1). Efficacy levels of both doses are assumed to increase sigmoidally, following Eq. 2. This approximates evidence that, for instance, the BNT162b2 vaccine (BioNTech/Pfizer) efficacy, compared to a placebo control, may not be observed until 12 days after being administered [20,21]. This is in contrast to recent work by [9] who used an asymptotic response function with a more rapid response in individual immunity. In the equations, $E_{1,i}$ and $E_{2,j}$ represent efficacies for the first and second dose of the vaccine on days $i$ (first dose) and $j$ (second dose), respectively.

$$E_{1,i} = \frac{V_{max,1}}{\left(1 + e^{\frac{\left(\frac{a}{2} - i\right)}{b}}\right)}$$

$$E_{2,j} = E_{1,a} + \frac{V_{max,2} - V_{max,1}}{\left(1 + e^{\frac{\left(\frac{a}{2} - j\right)}{b}}\right)} \tag{2}$$

The second dose was administered on the 22$^{\text{nd}}$ day after the individual received the first dose (parameter $a$ = 21 days). Additionally, the rate parameter was set to $b$ = 1.75, allowing for the second vaccine doses to meet the rise of the first dose, and $V_{max,k}$ represents the maximum efficacy for vaccine doses $k$ = 1 and $k$ = 2 (Table 1). The relationships for these efficacies are shown in Fig. 4, where values are tested for $0.4 \leq V_{max,1} \leq 0.75$ and $0.75 \leq V_{max,2} \leq 0.9$. The modeled responses for the four reference vaccines also are included, as is the single dose of the Johnson & Johnson vaccine (JJ = 0.72).

We also tested two different vaccination strategies. Both strategies select individuals from the S, E, or R classes that have not previously been vaccinated (see Fig. 1). Individuals that are infectious ($I$) are assumed to have been excluded from receiving a vaccine. The random strategy selects individuals for vaccination based on the appropriate probability for each simulation. The high degree strategy preferentially vaccinates individuals that have the most connections to neighbors.

## 3 Results

Model dynamics, using the tested parameter values in table 1, consistently exhibited an epidemic curve with a sigmoidally shaped number of recovered individuals over time (an example is shown in Fig 6). Of the 25,000 individuals, more individuals became infected on average when exposed to the more infectious strain ($R_0 = 2.5$ yielded 8,428 infected individuals while $R_0 = 5.0$ yielded 20,590 individuals, on average; F = 37,050; df = 1, 63,998; p < 0.001). Vaccination greatly reduced the number infected, with fewer subjects becoming infected with higher levels of vaccination rates and higher vaccine efficacies.

A sample simulation of a SARS-CoV-2 epidemic in a municipality with 25,000 individuals. Shown are the number of individuals that were susceptible ($S$), exposed ($E$), infectious ($I$), recovered ($R$), and the number of individuals vaccinated once ($V1$) and twice ($V2$). Random individuals were vaccinated with a first dose at a rate of 0.3% per day. These individuals were then given a second dose 21 days later. The simulation ended when there were no remaining exposed or infectious individuals ($E$ or $I$). For this simulation $R_0 = 5.0$, $k = 10$, $P = 0.01$,

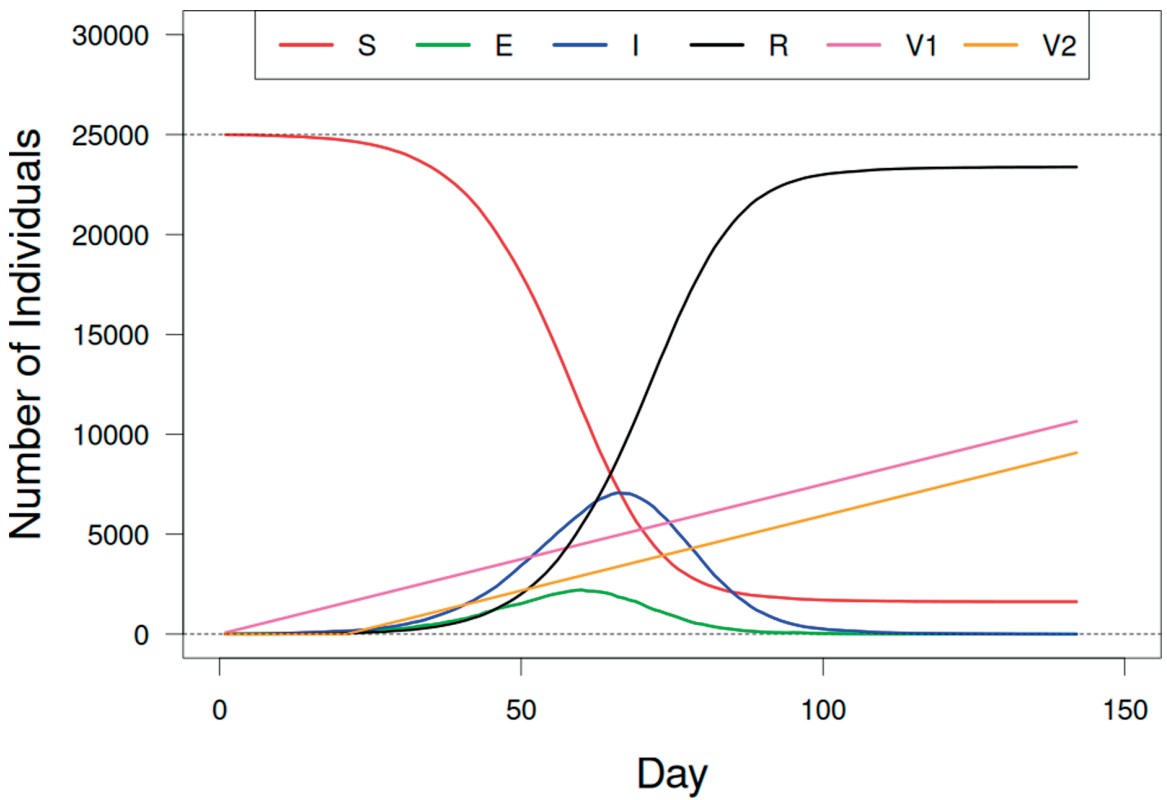

**Fig 6. Model simulation with vaccination.**

$V_{max,1} = 0.6$, and $V_{max,2} = 0.9$. In the end, 93.5% of the population became infected (entered the recovered class $R$).

The number of individuals infected depended most on three factors: the structure of the network ($P$), the percentage of the population vaccinated daily (% Vacc/Day), and the reproductive rate of the virus ($R_0$) (see Table 2). Additionally, these three factors, along with their interaction, were the most important factors in determining the response of the other three important response variables, including the duration of the epidemic, the number of individuals infectious at the epidemic peak, and day of the epidemic peak (Table 2). Therefore, the results strongly suggest that the two factors that we can control, the structure of the interaction network and the rate at which individuals become vaccinated, play a large role in determining the extent of an epidemic.

Fig. 7 provides a summary of the extent of how these three factors interacted to affect the number of individuals that became infected, epidemic duration, and the size and timing of the epidemic peaks (see effect sizes in Table 2). The number of individuals becoming infected was most affected by the virus' reproductive number ($R_0$). Interestingly, the vaccination rates were quite effective with an $R_0 = 2.5$ (upper-left panel, Fig 7). However, with an $R_0 = 5.0$ the virus was much more aggressive, especially at larger values of $P$ (greater mixing with shorter average path lengths), despite even high rates of vaccination (upper right graph of Fig 7).

Three-way interactions are shown for the four response variables (horizontally-paired graphs). These factors include network structure (P, see Fig 3), percent vaccinated per day (% Vacc/Day), and $R_0$. The interaction terms are all highly significant (see Table 2). For all simulations in this figure, vaccinated individuals received two doses. Error bars represent $\pm$ 95% confidence intervals and are present, but small, on most bars.

The duration of the epidemic was longest with low $R_0$ and no vaccination, as the virus agent made its way through nearly the entire population (second row panels in Fig 7). However, vaccination generally decreased the duration dramatically. The shortest duration occurred with $R_0 = 5.0$. Interestingly, at $R_0 = 5.0$ and relatively high network rewiring values ($P \geq 0.03$), increasing the rate of vaccination led to small *increases* in the duration of the epidemic. Duration, therefore, exhibited a complex response where short epidemic duration is not necessarily an indicator of successful disease containment.

The size of the epidemic peak, which indicates the potential number of individuals simultaneously seeking health care services, was highest with large $R_0$ (third row panels in Fig 7). For both levels of $R_0$ increasing vaccination rates decreased the epidemic peaks while larger values of $P$ (greater randomization of connections) led to increase epidemic sizes.

The timing of the epidemic peak also is important for health care providers as they prepare for the peak number of infected individuals to enter the health care system. Without vaccination on regularly structured networks led to peaks that occurred quite late in the epidemics

**Table 2. The percent sums of squares for the four response variables. The dominant factors influencing these responses were network structure ($P$), proportion of the population vaccinated per day (% Vacc/Day), $R_0$, and their three-way interaction (see Fig. 7). Note that the percentages of the variance explained are from the full, seven-factor ANOVAs completed for each effect. The greatest contributions to the overall effect are in bold text.**

| Response Variable | $P$ | % Vacc/Day | $R_0$ | $P$ x % Vacc/Day x $R_0$ | Total |
|---|---|---|---|---|---|
| Number Infected | 20.0 | 19.8 | **36.7** | 4.9 | 81.4 |
| Duration | 19.6 | 6.2 | 2.2 | **22.4** | 50.4 |
| Maximum Number Infectious | **45.9** | 5.1 | 32.5 | 0.9 | 84.4 |
| Day Maximum Infectious | 9.7 | 13.4 | 1.7 | **18.8** | 43.6 |

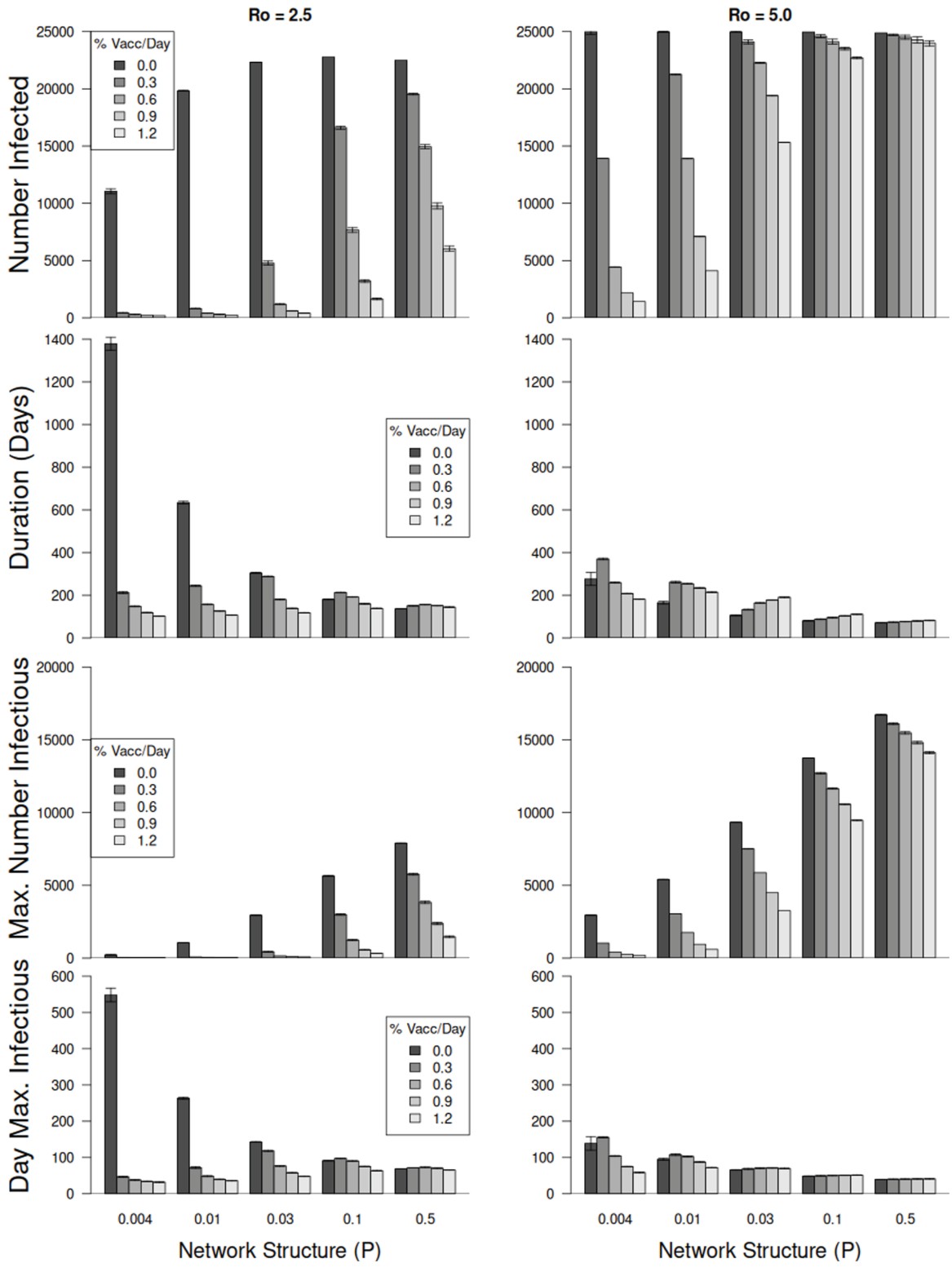

**Fig 7. Model response variables across three factors.**

(fourth row panels in Fig 7). This, however, is deceptive since those peaks were quite small. Also, this only occurred with $R_0 = 2.5$, not with the faster spreading virus. We see that the peak occurred soon into the epidemic with vaccination and with the faster spreading virus ($R_0 = 5.0$). This is particularly true for the faster spreading virus in a population that is more randomly structured (higher values of $P$).

### 3.1 The effect of different vaccines

Not surprisingly, the more effective vaccines resulted in fewer individuals becoming infected (Fig 8). More interestingly, administering a second dose resulted in little additional reduction in infections. This is likely due to the epidemic being quelled by the first vaccine and the virus having moved through the population by the time the second vaccine was able to take full effect (a total of 42 days after being administered). There was no significant interaction for the effect of administering either one or two vaccine doses on the number of individuals getting infected (Fig 8; F = 0.005; df = 21, 63,968; p = 1.0).

The number of individuals becoming infected by the variant with $R_0 = 2.5$ when receiving vaccines with different efficacies. Increasing the efficacy of the first dose reduces infections (F = 6.6 x $10^{10}$; df = 7, 63,968; p < 0.001) while increasing the efficacy of the second dose has little effect, although it is statistically significant (F = 3.9 x $10^9$; df = 3, 63,968; p < 0.001). There is no interaction between these factors (F = 9.8 x $10^6$; df = 21, 63,968; p = 1.0). Error bars represent $\pm$ 95% confidence intervals. Dashed, horizontal reference lines are at the lowest and highest means for clarity.

At a finer scale our model allows us to assess the efficacy of different vaccines (see Fig 4) as well as the efficacy of individuals receiving zero, one, or two vaccine doses. The tested vaccines had a wide range of efficacies but the effect of different vaccines on epidemic size was relatively small. Additionally, the first vaccine dose accounted for twenty times the effect of

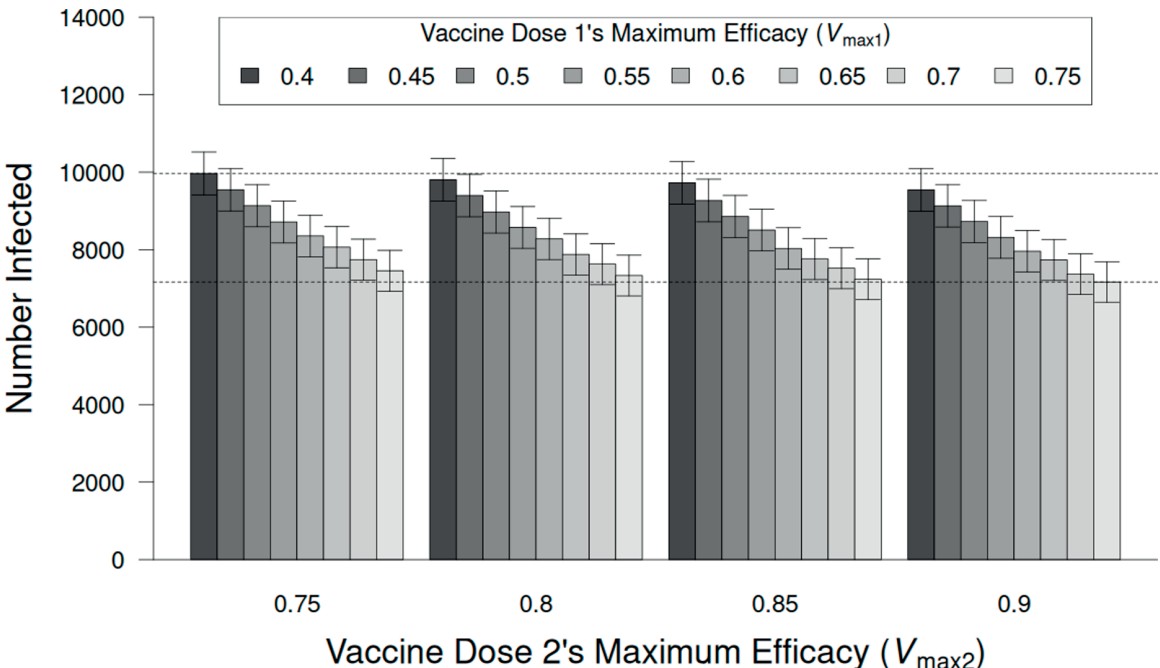

**Fig 8. The number of people infected after vaccination.**

adding the second dose, on average. Therefore, these results support that individuals should receive at least a single vaccine dose, even if the vaccine has a probability of 0.4 of protecting a person against infection.

The three-way interaction between the number of vaccine doses individuals received (one versus two), the percent of the population vaccinated per day (% Vacc/Day), and network structure (P) on the total number of individuals that got infected. These simulations represent the average for $R_0$ = 2.5 and 5.0. Each of the individual factors was significant, with network structure and number of doses per day being the most important factors. Note that the numerical differences between one and two doses without vaccination are not statistically different (e.g., for the rewire parameter of $P$ = 0.004; p > 0.999, Tukey HSD). Error bars represent $\pm$ 95% confidence intervals.

### The number of individuals infected depended on percent vaccinated daily, vaccine efficacy, and the number of doses administered

Vaccination reduced the number of infected individuals, such that vaccines with higher efficacy rates resulted in fewer cases (Fig 10). This effect was stronger in treatments with subjects receiving only a single dose (left graph, Fig 10). For tests that included individuals receiving a second dose, fewer individuals did become infected. However, the added benefit of the second dose, although statistically significant, did not result in large reduction in infections (right graph, Fig 10).

Number of infected individuals versus the vaccine efficacy when one (left panel) and two (right) vaccine doses were administered. The differences among the vaccine efficacies

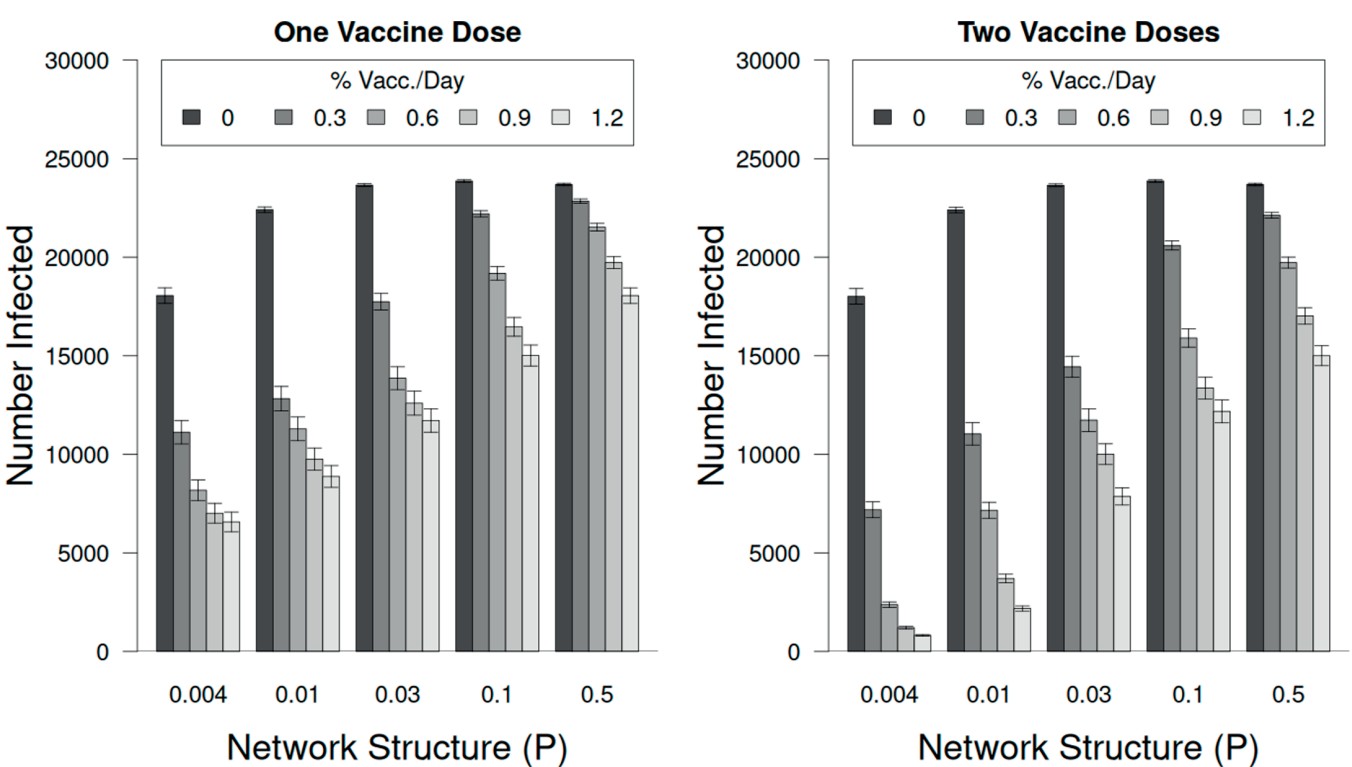

**Fig 9. The number of people infected depends on vaccine rates and structure but not number of doses.**

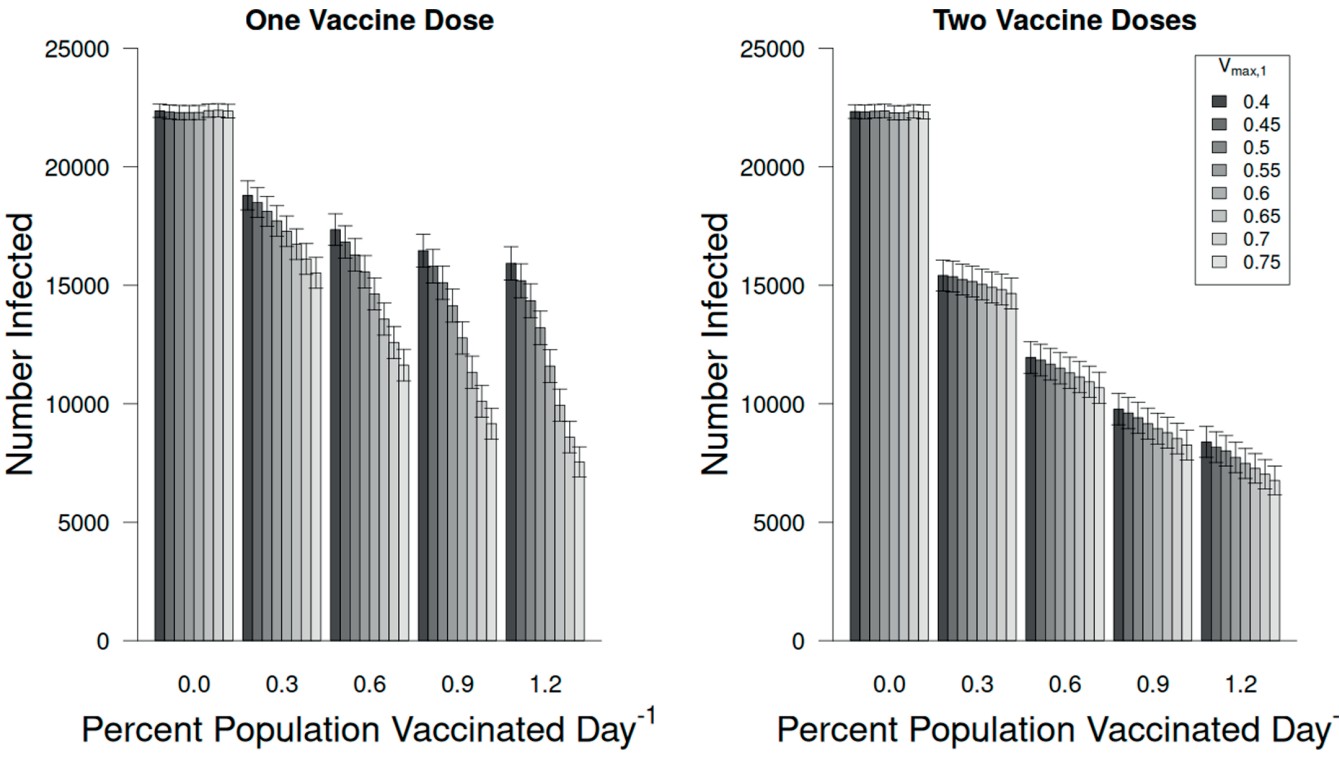

**Fig 10. Vaccine efficacy is more important for first dose than second dose.**

were quite important if individuals received only one dose but if individuals received two doses the differences were reduced. Note that these results average the response of individuals to the differences among the second dose. Error bars represent ± 95% confidence intervals.

### 3.2 The maximum number of infected individuals depended on vaccination, network structure, and $R_0$

As stated above the maximum number of individuals infected at the peak of the epidemic is critically important for health care service providers during an epidemic. Not surprisingly most individuals are infected without vaccination (Fig 11). Additionally, the maximum number of individuals infected at the peak is about twice as high with $R_0 = 5.0$ compared to $R_0 = 2.5$. However, when high rates of vaccination are implemented (two doses of vaccines with efficacies of $V_{max,1} = 0.75$ and $V_{max,2} = 0.9$) the number becoming infected with $R_0 = 2.5$ was found to be quite low. However, a strain of virus with an $R_0 = 5.0$ is able to break out quickly and cause widespread infections, particularly with well mixed populations ($P \geq 0.03$).

The maximum number of individuals infectious at the epidemic peak is sensitive to the viral reproduction rate ($R_0$). Without vaccination (left panel) the peaks increase in severity with increasing network connectivity ($P$), with peaks approximately doubling in height as $R_0$ increases from 2.5 to 5.0. With high rates of vaccination (1.2% day$^{-1}$) and high vaccine efficacy ($V_{max,1} = 0.75$, $V_{max,2} = 0.9$) the peaks are much lower with $R_0 = 2.5$ (right panel). However, as seen in Fig 9, when $R_0 = 5.0$ even high vaccination rates fail to contain the epidemic peaks

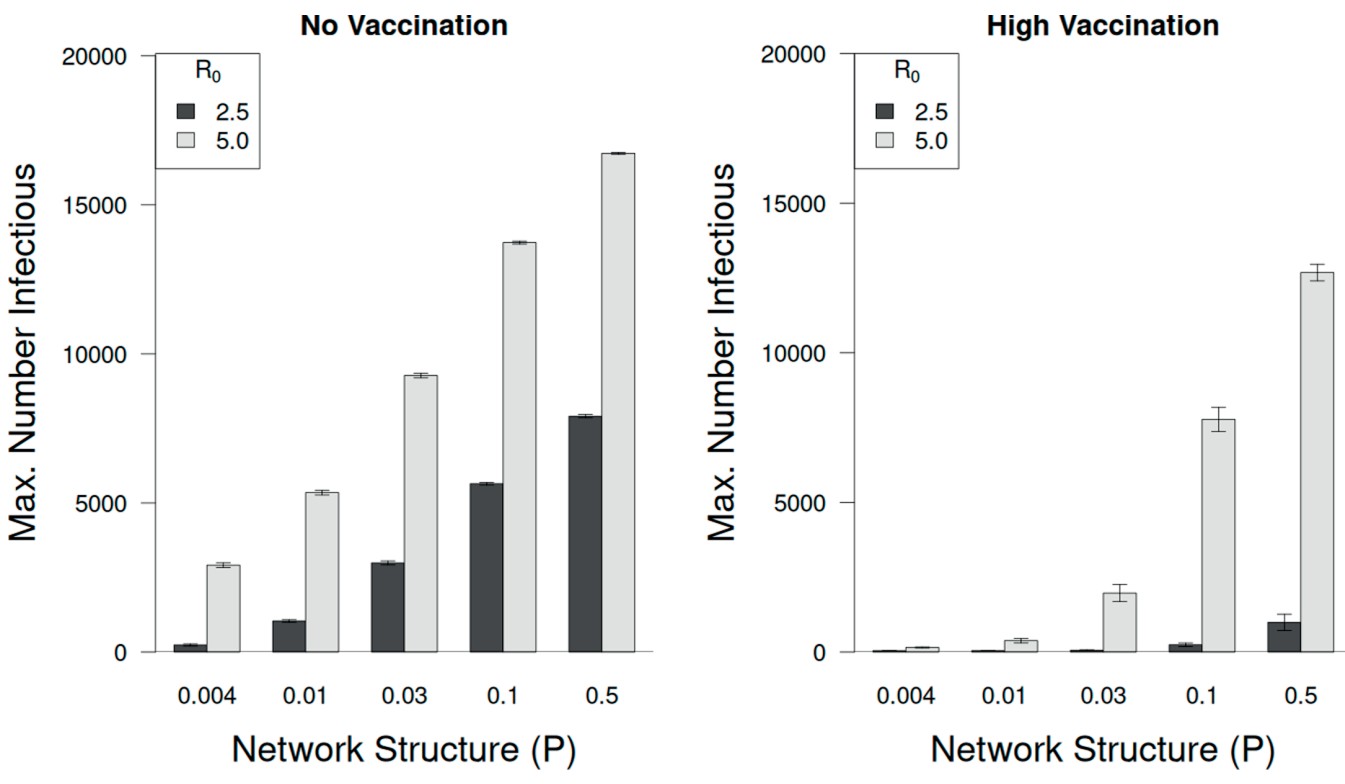

**Fig 11. Peak number infectious depends on $R_0$, vaccination, and network structure.**

when the network connectivity is high (right panel). Error bars represent $\pm$ 95% confidence intervals.

## Vaccination strategy and network structure affected the number of individuals infected

The two tested strategies differed significantly with preferential vaccination of more connected individuals (high degree) outperforming a random strategy, on average (F = 4,932; df = 1, 5,040; p < 0.001; see Fig. 12). Not surprisingly, the vaccination strategy interacted significantly with the structure of the network (F = 1,435; df = 3, 5,040; p < 0.001) because with $P>0.0$ led to individuals with relatively high degrees that were then selected for vaccination, functionally reducing more edges. As can be seen in Fig 3, the opportunity to preferentially vaccinate individuals of higher degree increases as the degree distributions broaden. For simulations with $R_0$ = 2.5 differences between the vaccination strategies was seen only with greater randomness in the network structure ($P \geq 0.1$).

The number of individuals getting infected was sensitive to vaccination strategies but only under certain circumstances. With an ($R_0$ = 2.5) the hub (high degree) vaccination strategy was noticeably more effective as the degree distribution broadened (large $P$). However, this strategy had little effect on controlling the epidemic when $R_0$ = 5.0. Interestingly, the hub strategy was more effective on more regular networks when $R_0$ = 5.0 ($P \leq 0.03$). The data here are from simulations with only the most effective vaccines administered ($V_{max,1}$ = 0.75, $V_{max,2}$ = 0.9). Error bars represent $\pm$ 95% confidence intervals.

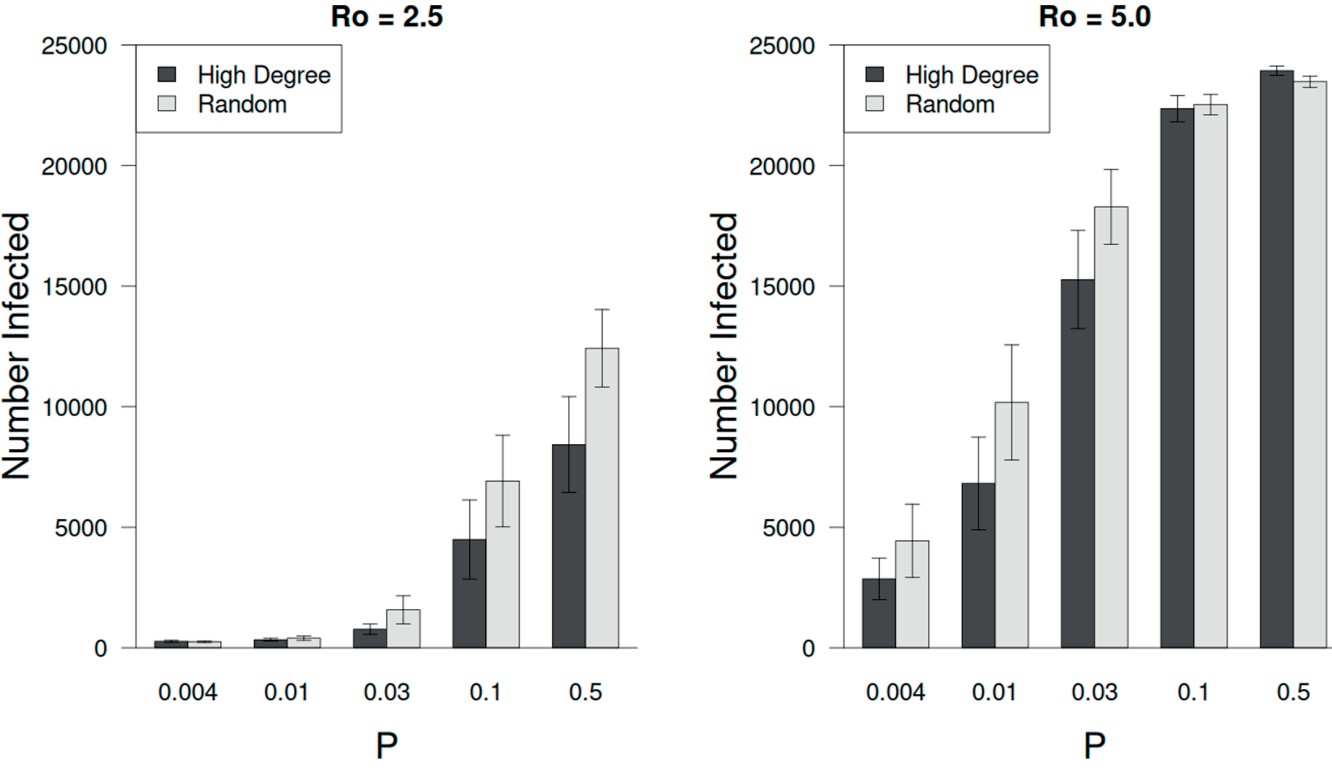

**Fig 12. Vaccination strategy based on degree improves vaccine efficacy.**

## 4 Discussion

This paper describes the dynamics of simulated SARS-CoV-2 spreading through a network population of 25,000 individuals. We test the effects of employing one or two doses of vaccines with different efficacies, and administered at different rates, on reducing the size of the epidemic. We also investigate the effect of different small-world network structures on epidemic development, from a nearly regular network to networks approximating a type of random structure while maintain a constant number of edges connecting individuals.

Our model yields dynamics that follow a standard SEIR curve, even when vaccinations were administered (see Fig 6). Since the population is limited to just 25,000 individuals the epidemics that developed only last up to approximately six months. The benefit of this individual-based model is that we're able to assess various factors on the number of individuals infected, epidemic duration, and the timing and size of the epidemic peak. This networked population shows that this disease agent is capable of spreading to a majority of individuals with and without interventions (see top row, Fig 7).

Overall, the tested values for the basic reproductive rate of the virus ($R_0$ = 2.5 or 5.0) represent a relatively conservative range for the early part of the original 2020 epidemic, with median estimates across different countries ranging from 3.5 to 5.9 [14]. Our tested values of $R_0$ were a bit lower than found in real populations, yet still resulted in extensive epidemics.

Additionally, we found that network structure and vaccination rate were quite important in affecting epidemic size and that they interacted statistically with $R_0$. These significant effects were seen in the number of individuals infected, epidemic duration, and epidemic peak size

and peak timing (see Table 2 and Fig 7). These response variables are of great concern during epidemics for general health concerns but also for health systems and providers because of the potential of high patient demand to become overwhelming for service providers. Other approaches have been investigated that also would help to reduce epidemic effects, such as increasing primary health care interventions [22], but these were not investigated.

We were able to test the effect of an array of different vaccines which overlapped with known vaccine efficacies. These were provided in either one or two doses, which resulted in statistically different levels of efficacy in protecting individuals. Our model assumes that the vaccines are, at best, only 90% effective at preventing a person from becoming infectious themselves ($VE_2$), assuming they come into contact with an infectious neighbor. Although the effect of different vaccines was significant the effect is relatively small, compared to the difference between simply vaccinating and not vaccinating individuals. There is evidence, however, that vaccinated individuals became infected early in the original COVID-19 epidemic (with the early Delta variant, B.1.617.2), a process referred to as "breakthrough infections," and that even some of those individuals could become infectious [23]. This effect was similar to the effect in this model of vaccinated individuals not gaining protection and, therefore, becoming infected and infectious.

The model suggests that, for the total number of infections, the structure of the network is similar in importance to the proportion of the population getting vaccinated. Therefore, efforts to reduce the spread of the virus should include the sharing of information about connectivity in social networks as well as the importance of getting the latest vaccines. Additionally, who receives the vaccines is important, as was seen in the difference between random and degree-based vaccination strategies, with the latter being significantly more effective.

The model does not investigate the effect of social distancing or quarantining of exposed/infectious individuals, although these, and other, non-pharmaceutical interventions have been found to be helpful in reducing the size and spread rate of epidemics [24,25,27]. There appears to remain some controversy over the two meter social distancing rule that many countries adopted to minimize transmission of the virus between individuals. However, there is support that some level of social distancing does reduce transmission rate [28].

As the network structure is changed from nearly regular to more random (larger values of the rewiring parameter $P$) both the average path length and the clustering coefficient decrease so that infected individuals will more likely have susceptible neighbors allowing the disease agent to spread further through the population (top row of Fig. 7). The structure of the networks used in this paper exhibit the characteristic of having relatively high clustering coefficients and relatively low average path lengths, similar to network examples provided in Watts and Strogatz [15] and, for example, a college network [25].

As mentioned, the strategy used to pick individuals to get vaccinated is important. We chose to investigate choosing either individuals randomly or preferentially based on degree (hub vaccination strategy). The latter strategy was significantly more effective, which has been found in other studies (e.g., [17,26]). However, we also found that vaccination strategy interacts significantly with network structure and viral basic reproductive number ($R_0$) (Fig 12).

Interestingly, the duration of epidemics increased with increasing vaccination. This can be explained by the increase in path lengths among the susceptible individuals within the network. Using this model we were able to assess this and found that the high-degree vaccination strategy was more effective at reducing the duration of epidemics. This also was true for tests when only one dose was administered, although more individuals were infected and the duration of the epidemics were somewhat longer (Fig 7). In tests where a second vaccination dose was administered, the added effectiveness was significant, although the effect size was

small. This is likely due to individuals receiving the second dose too late in the epidemic (see Fig 10).

Our model assumes that all individuals are equally susceptible to SARS-CoV-2 infection and that there were no negative outcomes from vaccination. We also ignore any acute or chronic effects due to SARS-CoV-2 infection. Our model does not allow for breakthrough infections (infections occurring in vaccinated individuals), although the number of such occurrences in our population of 25,000 would be low [29]. However, vaccinations in our model never provide 100% coverage, either (see Fig 4). Finally, this model does not address the ability of SARS-CoV-2 to evolve or for people to change their behavior that likely occurs as an epidemic develops [30].

## 5 Data analysis

All analyses and model runs were completed using R [31]. Networks were constructed using the igraph package (igraph.org).

## 6 Acknowledgments

The authors would like to thank Marisa Presutto, Grace Maley, and Christopher Leary for their helpful insights. Additionally, we thank the R [31] and RStudio [32]. developers.

## Author contributions

**Conceptualization:** Gregg Hartvigsen, Yannis Dimitroff.

**Data curation:** Gregg Hartvigsen.

**Formal analysis:** Gregg Hartvigsen.

**Investigation:** Yannis Dimitroff.

**Methodology:** Gregg Hartvigsen.

**Project administration:** Gregg Hartvigsen.

**Resources:** Gregg Hartvigsen.

**Software:** Gregg Hartvigsen, Yannis Dimitroff.

**Supervision:** Gregg Hartvigsen.

**Validation:** Gregg Hartvigsen.

**Visualization:** Gregg Hartvigsen, Yannis Dimitroff.

**Writing – original draft:** Gregg Hartvigsen, Yannis Dimitroff.

**Writing – review & editing:** Gregg Hartvigsen, Yannis Dimitroff.

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
