## [Decision Letter · Decision Letter 0]

10 Apr 2025

PONE-D-25-01005Modeling the SARS-CoV-2 Epidemic and the Efficacy of Different Vaccines Across Different Network StructuresPLOS ONE

Dear Dr. Hartvigsen,

Thank you for submitting your manuscript to PLOS ONE. After careful consideration, we feel that it has merit but does not fully meet PLOS ONE’s publication criteria as it currently stands. Therefore, we invite you to submit a revised version of the manuscript that addresses the points raised during the review process.

**As you will see in the attached reviewers' reports, both reviewers have suggested that the manuscript would benefit from additional details and clearer explanations, particularly in certain sections. Enhancing the clarity and depth of these parts will help strengthen the overall impact and readability of your work.**

**Please respond to each reviewer comment individually, and make sure that all revisions are clearly indicated in the manuscript (e.g., using tracked changes or highlighting). If you disagree with any specific point, you are welcome to explain your reasoning in your response letter.**

We look forward to receiving your revised manuscript.

Kind regards,

David Moriña Soler

Academic Editor

PLOS ONE

**Journal Requirements:**

1. When submitting your revision, we need you to address these additional requirements. Please ensure that your manuscript meets PLOS ONE's style requirements, including those for file naming. The PLOS ONE style templates can be found at https://journals.plos.org/plosone/s/file?id=wjVg/PLOSOne_formatting_sample_main_body.pdf and https://journals.plos.org/plosone/s/file?id=ba62/PLOSOne_formatting_sample_title_authors_affiliations.pdf 2. Please note that PLOS ONE has specific guidelines on code sharing for submissions in which author-generated code underpins the findings in the manuscript. In these cases, we expect all author-generated code to be made available without restrictions upon publication of the work. Please review our guidelines at https://journals.plos.org/plosone/s/materials-and-software-sharing#loc-sharing-code and ensure that your code is shared in a way that follows best practice and facilitates reproducibility and reuse. 3. Thank you for stating the following in the Acknowledgments Section of your manuscript: We appreciate the discussions with Chris Leary, Grace Maley, and Marisa Presutto and the support of the Sponsored Programs at SUNY College at Geneseo, NY. We note that you have provided funding information that is not currently declared in your Funding Statement. However, funding information should not appear in the Acknowledgments section or other areas of your manuscript. We will only publish funding information present in the Funding Statement section of the online submission form. Please remove any funding-related text from the manuscript and let us know how you would like to update your Funding Statement. Currently, your Funding Statement reads as follows: The author(s) received no specific funding for this work. Please include your amended statements within your cover letter; we will change the online submission form on your behalf. 4. Thank you for uploading your study's underlying data set. Unfortunately, the repository you have noted in your Data Availability statement does not qualify as an acceptable data repository according to PLOS's standards. At this time, please upload the minimal data set necessary to replicate your study's findings to a stable, public repository (such as figshare or Dryad) and provide us with the relevant URLs, DOIs, or accession numbers that may be used to access these data. For a list of recommended repositories and additional information on PLOS standards for data deposition, please see https://journals.plos.org/plosone/s/recommended-repositories. 5. Please amend either the abstract on the online submission form (via Edit Submission) or the abstract in the manuscript so that they are identical.

Reviewers' comments:

Reviewer's Responses to Questions

**Comments to the Author**

1. Is the manuscript technically sound, and do the data support the conclusions?

Reviewer #1: Yes

Reviewer #2: Yes

2. Has the statistical analysis been performed appropriately and rigorously? 

Reviewer #1: Yes

Reviewer #2: Yes

3. Have the authors made all data underlying the findings in their manuscript fully available?

Reviewer #1: Yes

Reviewer #2: Yes

4. Is the manuscript presented in an intelligible fashion and written in standard English?

Reviewer #1: Yes

Reviewer #2: Yes

5. Review Comments to the Author

**Reviewer #1:** The original research article come with an interesting study of model of vaccine efficacy using the network modeling. I really enjoyed the reading the paper the method provides a solid overview of the model's construction, network structure, and vaccination strategies, however, I am not such expert in the field and some further explanation will be nice to better understanding the paper. For example, the network parameterization (P) - was chosen from 0,004 to 0,5. Why the selected value were chosed, and how does the choice of P relate to the epidemic patterns? Also I am not very familiar with 'vaccine strategies', which was selected as random or high degree. There are few words to this point (line 164-165), but please add more explanation. In the result part, starting L265 - the paragraph is for my hard to understand, I would like to read some more explanation regarding why a second dose has less effect. If I understand it correctly, the virus has already spread significantly by the time the second dose takes effect, which could clarify why the impact of the second dose is smaller. But this is somehow not clear explained in the text, or maybe I understand it wrong?

**Reviewer #2:** I think the paper is well-written, and the work is both interesting and quite comprehensive. Below are a few suggestions and clarifications that might enhance the manuscript:

- In section 3, Discussion, the authors mention that the reproductive rate of SARS-CoV-2 ranged from 3.5 to 5.9, depending on the country. It might be helpful to also reference this information in Section 1, Methods, to provide a stronger rationale for choosing $R_0 = 2.5$ and $R_0 = 5.0$.

- In equation (2), $b = 1.75$ is used. A brief explanation of how this value was determined would be valuable for clarity.

- On line 182, in the caption of Figure 6, I believe the term "infected" should be replaced with "infectious".

- While reading the paper, I found myself wondering if there is a method to estimate the network structure of a real population with a reasonable degree of accuracy. It may be helpful to include a brief discussion on how the simulations could be linked to real-world scenarios in this regard.

- It appears that the section on Acknowledgments is repeated.

6. PLOS authors have the option to publish the peer review history of their article (what does this mean?). If published, this will include your full peer review and any attached files.

Reviewer #1: **Yes: **Elena Bencurova

Reviewer #2: No

---

## [Author Response · Author response to Decision Letter 1]

22 Apr 2025

See attached document: "Response to Reviewers.pdf"

---

## [Decision Letter · Decision Letter 1]

7 May 2025

Modeling the SARS-CoV-2 Epidemic and the Efficacy of Different Vaccines Across Different Network Structures

PONE-D-25-01005R1

Dear Dr. Hartvigsen,

We’re pleased to inform you that your manuscript has been judged scientifically suitable for publication and will be formally accepted for publication once it meets all outstanding technical requirements.

Kind regards,

David Moriña Soler

Academic Editor

PLOS ONE

Reviewers' comments:

Reviewer's Responses to Questions

**Comments to the Author**

1. If the authors have adequately addressed your comments raised in a previous round of review and you feel that this manuscript is now acceptable for publication, you may indicate that here to bypass the “Comments to the Author” section, enter your conflict of interest statement in the “Confidential to Editor” section, and submit your "Accept" recommendation.

Reviewer #2: All comments have been addressed

2. Is the manuscript technically sound, and do the data support the conclusions?

Reviewer #2: Yes

3. Has the statistical analysis been performed appropriately and rigorously? 

Reviewer #2: Yes

4. Have the authors made all data underlying the findings in their manuscript fully available?

Reviewer #2: Yes

5. Is the manuscript presented in an intelligible fashion and written in standard English?

Reviewer #2: Yes

6. Review Comments to the Author

Reviewer #2: (No Response)

7. PLOS authors have the option to publish the peer review history of their article (what does this mean?). If published, this will include your full peer review and any attached files.

Reviewer #2: No

---

## [Editor Report · Acceptance letter]

PONE-D-25-01005R1

PLOS ONE

Dear Dr. Hartvigsen,

I'm pleased to inform you that your manuscript has been deemed suitable for publication in PLOS ONE. Congratulations! Your manuscript is now being handed over to our production team.

Kind regards,

on behalf of

Dr. David Moriña Soler

Academic Editor

PLOS ONE